# PD-1/PD-L1 Inhibitors plus Chemotherapy Versus Chemotherapy Alone for Resectable Non-Small Cell Lung Cancer: A Systematic Review and Meta-Analysis of Randomized Controlled Trials

**DOI:** 10.3390/cancers15215143

**Published:** 2023-10-26

**Authors:** Eric Pasqualotto, Francisco Cezar Aquino de Moraes, Matheus Pedrotti Chavez, Maria Eduarda Cavalcanti Souza, Anna Luíza Soares de Oliveira Rodrigues, Rafael Oliva Morgado Ferreira, Lucca Moreira Lopes, Artur Menegaz de Almeida, Marianne Rodrigues Fernandes, Ney Pereira Carneiro dos Santos

**Affiliations:** 1Department of Medicine, Federal University of Santa Catarina, Florianópolis 88040-900, Santa Catarina, Brazil; eric.pasqualotto@grad.ufsc.br (E.P.); matheus.pedrotti@grad.ufsc.br (M.P.C.); rafael.morgado@grad.ufsc.br (R.O.M.F.); 2Oncology Research Center, Federal University of Pará, Belém 66073-005, Pará, Brazil; npcsantos.ufpa@gmail.com (N.P.C.d.S.); fernandesmr@yahoo.com.br (M.R.F.); 3Department of Medicine, University of Pernambuco, Recife 50670-901, Pernambuco, Brazil; eduarda.csouza@upe.br; 4Department of Medicine, University Center of João Pessoa, João Pessoa 58053-000, Paraíba, Brazil; annaluizasor@gmail.com; 5Independent Researcher, São Paulo 09090-720, São Paulo, Brazil; luccausmle2022@gmail.com; 6Department of Medicine, Federal University of Mato Grosso, Sinop 78550-704, Mato Grosso, Brazil; artur.almeida@sou.ufmt.br

**Keywords:** immune checkpoint inhibitors, neoadjuvant therapy, lung cancer, immunotherapy, early stage

## Abstract

**Simple Summary:**

Immunotherapy, particularly programmed cell death protein 1 (PD-1) and programmed death-ligand 1 (PD-L1) inhibitors, has been responsible for changing the natural history of advanced or metastatic non-small cell lung cancer. However, its use in the resectable stage is not yet fully elucidated. Therefore, we aimed to evaluate the efficacy and safety of neoadjuvant and adjuvant use of PD-1/PD-L1 inhibitors plus chemotherapy versus chemotherapy alone in resectable stage (I-III) non-small cell lung cancer. Our findings suggest that the incorporation of PD-1/PD-L1 inhibitors alongside chemotherapy offers a promising prospect for reshaping the established treatment paradigms for patients diagnosed with resectable stages of non-small cell lung cancer.

**Abstract:**

**Background:** The benefit of adding programmed cell death protein 1 (PD-1)/programmed death-ligand 1 (PD-L1) inhibitors to the treatment of early-stage non-small cell lung cancer (NSCLC), both neoadjuvant therapy (NAT) and adjuvant therapy (AT), is not yet fully elucidated. **Methods:** We searched PubMed, Embase, and Cochrane databases for randomized controlled trials (RCT) that investigated PD-1/PD-L1 inhibitors plus chemotherapy for resectable stage NSCLC. We computed hazard ratios (HRs) or odds ratios (ORs) for binary endpoints, with 95% confidence intervals (CIs). **Results:** A total of seven RCTs comprising 3915 patients with resectable stage NSCLC were randomized to chemotherapy with or without PD-1/PD-L1 inhibitors as NAT or AT. As NAT, the PD-1/PD-L1 inhibitors plus chemotherapy group demonstrated significantly improved overall survival (HR 0.66; 95% CI 0.51–0.86) and event-free survival (HR 0.53; 95% CI 0.43–0.67) compared with the chemotherapy alone group. There was a significant increase in favor of the PD-1/PD-L1 inhibitors plus chemotherapy group for major pathological response (OR 6.40; 95% CI 3.86–10.61) and pathological complete response (OR 8.82; 95% CI 4.51–17.26). Meanwhile, as AT, disease-free survival was significant in favor of the PD-1/PD-L1 inhibitors plus chemotherapy group (HR 0.78; 95% CI 0.69–0.90). **Conclusions:** In this comprehensive systematic review and meta-analysis of RCTs, the incorporation of PD-1/PD-L1 inhibitors alongside chemotherapy offers a promising prospect for reshaping the established treatment paradigms for patients diagnosed with resectable stages of NSCLC. Moreover, our analyses support that neoadjuvant administration with these agents should be encouraged, in light of the fact that it was associated with an increased survival and pathological response, at the expense of a manageable safety profile.

## 1. Introduction

Non-small cell lung cancer (NSCLC) is the most common lung cancer, accounting for approximately 80–85% of all cases [1,2]. In about 50% of cases, the disease is either localized (stages I and II) or locally advanced (stage III) [3,4]. The standard treatment for stage I and II NSCLC, as well as specific IIIA cases, involves surgical resection. In this scenario, the 5-year survival rate for patients with stage I-II NSCLC remains at 92%; however, it drops to 36% for patients with stage IIIA [5,6,7].

Immune checkpoint inhibitors (ICIs), particularly antibodies to programmed cell death protein 1 (anti-PD1) and programmed death ligand 1 (anti-PD-L1), are used in NSCLC with the rationale that blocking programmed cell death protein 1 (PD-1) on activated T cells and programmed death-ligand 1 (PD-L1) on tumor cells could reinvigorate cytotoxic TCD8+ cells by activating host adaptive immunity [8,9]. In NSCLC, the use of anti-PD-1/PD-L1 agents has demonstrated improved overall survival (OS) and progression-free survival (PFS) following chemoradiotherapy in unresectable stage III disease. As a result, these agents have been approved for the treatment of advanced or metastatic NSCLC in cases without molecular alterations [10,11,12,13,14,15,16,17,18]. 

In the phase III clinical trial CheckMate 816, administration of Nivolumab together with chemotherapy as neoadjuvant therapy (NAT) improved event-free survival (EFS) compared with chemotherapy alone [19]. Additionally, in IMpower010 trial, adjuvant Atezolizumab (PD-L1 inhibitor) plus chemotherapy demonstrated a benefit for the risk of recurrence or death compared to the best supportive care for NSCLC II-IIIA, in patients with at least 1% PD-L1 expression [20]. 

Therefore, in this systematic review and meta-analysis of randomized controlled trials (RCTs), we aim to clarify the efficacy and safety of using neoadjuvant or adjuvant PD-1/PD-L1 inhibitors plus chemotherapy versus chemotherapy alone in resectable stage (I-III) NSCLC.

## 2. Methods

### 2.1. Protocol and Registration

This systematic review followed the Preferred Reporting Items for Systematic Reviews and Meta-Analysis (PRISMA) guidelines [21]. The protocol was registered in the International Prospective Register of Systematic Reviews (PROSPERO) with registration number CRD42023447777.

### 2.2. Eligibility Criteria

Studies that met the following eligibility criteria were included: (1) RCT; (2) comparison of neoadjuvant or adjuvant PD-1/PD-L1 inhibitors plus chemotherapy versus chemotherapy; (3) adult patients with early stage I-III NSCLC (according American Joint Committee on Cancer, 7th edition); (4) complete surgical resection including negative margins in studies with adjuvant therapy (AT); (5) no previous anti-cancer therapy in studies with NAT; and (6) Eastern Cooperative Oncology Group (ECOG) performance status score of 0, 1, or 2 (on a 5-point scale in which higher scores reflect greater disability). We excluded studies (1) with overlapping populations; (2) without outcomes of interest; and (3) with unpublished complete results. Inclusion and exclusion criteria for the RCTs included in this systematic review and meta-analysis are detailed in Appendix A.

### 2.3. Search Strategy and Data Extraction

PubMed, Cochrane Library, and Embase were systematically searched on 4 August 2023. The search strategy is detailed in Appendix A. In addition, backwards snowballing was performed, aimed at the inclusion of additional studies. Those found in the databases and in the references of the articles were incorporated into the reference management software (EndNote^®^, version X7, Thomson Reuters, Philadelphia, PA, USA). Duplicate articles were removed, using both automated and manual methods. Subsequently, two reviewers (E.P. and L.M.L.) independently analyzed the titles and abstracts of the identified articles. Disagreements were resolved by consensus between the two authors and senior author (E.P., L.M.L., and N.P.C.d.S.)

The following baseline characteristics were extracted: (1) ClinicalTrials.gov, accessed on 28 August 2023, Identifier and study design; (2) number of patients allocated for each arm; (3) regimen details in experimental and control arm; and (4) main patient’s characteristics. Two authors (A.L.S.O.R and M.E.C.S) collected pre-specified baseline characteristics and outcome data.

### 2.4. Endpoints

Outcomes of interest were (1) OS; (2) disease-free survival (DFS); (3) EFS; (4) major pathological response (MPR); (5) pathological complete response (pCR); patients with any grade of (6) fatigue; (7) pruritus; (8) arthralgia; (9) diarrhea; (10) increased alanine aminotransferase; (11) hypothyroidism; (12) nausea; (13) rash; (14) decreased appetite; (15) anemia; (16) constipation; (17) decreased neutrophil count; patients with grade ≥ 3 of (18) fatigue; (19) diarrhea; (20) increased alanine aminotransferase; (21) decreased neutrophil count; (22) rash; and (23) decreased appetite.

We defined (1) OS, as the period from randomization to all-cause mortality; (2) DFS, as the time from randomization to the occurrence of loco-regional or metastatic recurrence, appearance of a second primary of NSCLC or other malignancy, or death from any cause, whichever occurred first; (3) EFS, as the interval between randomization and any disease progression that would render the patient ineligible for surgery, disease progression or recurrence after surgery, disease progression in the absence of surgery, presence of unresectable tumor, or mortality from any cause. Regarding response, we defined (1) MPR as the presence of ≤10% residual viable tumor cells in the primary tumor and in the sampled lymph nodes; (2) pCR was determined by the complete absence of viable tumor cells at the primary tumor site and in the surgically removed lymph nodes after NAT.

### 2.5. Risk of Bias Assessment

The quality assessment of individual RCTs was carried out using the Cochrane Collaboration tool for assessing risk of bias in randomized trials (RoB 2) [22]. Two authors (E.P. and R.M.O.F.) independently conducted the assessment, and disagreements were resolved by consensus. For each trial, a risk of bias score was assigned, indicating whether it was at a high, low, or unclear risk of bias across five domains: randomization process, deviations from intended interventions, missing outcomes, measurement of outcomes, and selection of reported results. To examine publication bias, contour-enhanced funnel plots [23] were visually inspected and assessed by Egger’s regression asymmetry [24] and Begg’s rank correlation test [25]. 

### 2.6. Sensitivity Analyses

#### 2.6.1. Subgroup Analyses

Subgroup analyses included data restricted to (1) NAT and (2) AT.

#### 2.6.2. Dominant Studies

Leave-one-out procedures were used to identify influential studies and their effect on the pooled estimates, evaluating the heterogeneity. This procedure was carried out removing data from one study and reanalyzing the remaining data, confirming that the pooled effect sizes did not result from single-study dominance.

### 2.7. Statistical Analysis

Binary endpoints were evaluated with hazard ratios (HRs) or odds ratios (ORs), with 95% confidence intervals (CIs). The Cochrane *Q*-test and I^2^ statistics were used to assess heterogeneity; *p* values > 0.10 and I^2^ values > 25% were considered to indicate significance for heterogeneity [26]. We used DerSimonian and Laird random-effect models for all endpoints [27]. Statistical analyses were performed using R statistical software, version 4.2.3 (R Foundation for Statistical Computing).

## 3. Results

### 3.1. Study Selection and Characteristics

The initial search yielded 6454 results, as detailed in Figure 1. After the removal of duplicate records, and the assessment of the studies based on title and abstract, 93 full-text studies remained for full review according to prespecified criteria. Of these, seven RCTs were included comprising 3915 patients [19,20,28,29,30,31,32]. A total of 1975 patients with NSCLC were randomized to PD-1/PD-L1 inhibitors plus chemotherapy, while 1940 received chemotherapy alone. A total of 1733 patients received NAT and 2182 patients received AT. The follow-up period ranged from 14.1 to 35.5 months. The median age ranged from 61.0 to 65.0 years. A total of 223 patients had epidermal growth factor receptor (EGFR) mutation, while 68 had anaplastic lymphoma kinase (ALK) mutation. Study and participant characteristics are detailed in Table 1 and Appendix A. Different treatment regimes were carried out in the included RCT involving chemotherapy and PD-1/PD-L1 inhibitors; more details are presented in Appendix A.

### 3.2. Pooled Analysis of All Studies

#### 3.2.1. Overall survival

The PD-1/PD-L1 inhibitors plus chemotherapy group showed no significant difference compared to the chemotherapy alone group for OS (HR 0.80; 95% CI 0.64–1.01; *p* = 0.062; I^2^ = 47%; Figure 2). 

#### 3.2.2. Neoadjuvant Therapy

The PD-1/PD-L1 inhibitors plus chemotherapy group demonstrated significantly improved OS (HR 0.66; 95% CI 0.51–0.86; *p* < 0.01; I^2^ = 0%; Figure 2) and EFS (HR 0.53; 95% CI 0.43–0.67; *p* < 0.01; I^2^ = 20%; Figure 3) compared with the chemotherapy alone group.

There was an increase in favor of the PD-1/PD-L1 inhibitors plus chemotherapy group for MPR (OR 6.40; 95% CI 3.86–10.61; *p* < 0.01; I^2^ = 66%; Figure 4B) and pCR (OR 8.82; 95% CI 4.51–17.26; *p* < 0.01; I^2^ = 48%; Figure 4C).

#### 3.2.3. Adjuvant Therapy

There was no significant difference between groups for OS (HR 0.96; 95% CI 0.78–1.17; *p* = 0.687; I^2^ = 6%; Figure 2). The estimated DFS was significant in favor of the PD-1/PD-L1 inhibitors plus chemotherapy group (HR 0.78; 95% CI 0.69–0.90; *p* < 0.01; I^2^ = 0%; Figure 3).

#### 3.2.4. Adverse Events

There was a significant increase in the PD-1/PD-L1 inhibitors plus chemotherapy group for any grade of arthralgia (OR 1.65; 95% CI 1.27–2.14; *p* < 0.01; I^2^ = 0%; Appendix A), increased alanine aminotransferase (OR 2.01; 95% CI 1.19–3.40; *p* < 0.01; I^2^ = 70%; Appendix A), hypothyroidism (OR 6.77; 95% CI 4.10–11.21; *p* < 0.01; I^2^ = 21%; Appendix A), and rash (OR 2.26; 95% CI 1.34–3.80; *p* < 0.01; I^2^ = 37%; Appendix A). 

There was no significant difference between groups for any grade of fatigue (OR 1.19; 95% CI 0.96–1.48; *p* = 0.11; I^2^ = 0%; Appendix A), pruritus (OR 4.01; 95% CI 0.80–20.00; *p* = 0.09; I^2^ = 87%; Appendix A), diarrhea (OR 1.16; 95% CI 0.80–1.67; *p* = 0.44; I^2^ = 31%; Appendix A), nausea (OR 1.20; 95% CI 0.51–2.81; *p* = 0.68; I^2^ = 85%; Appendix A), decreased appetite (OR 1.07; 95% CI 0.72–1.57; *p* = 0.74; I^2^ = 55%; Appendix A), anemia (OR 0.94; 95% CI 0.71–1.26; *p* = 0.68; I^2^ = 14%; Appendix A), constipation (OR 1.01; 95% CI 0.80–1.28; *p* = 0.91; I^2^ = 0%; Appendix A), and decreased neutrophil count (OR 0.76; 95% CI 0.48–1.19; *p* = 0.23; I^2^ = 55%; Appendix A).

In addition, there was no significant difference between groups for grade ≥ 3 of fatigue (OR 1.19; 95% CI 0.41–3.46; *p* = 0.75; I^2^ = 0%; Appendix A), diarrhea (OR 2.60; 95% CI 0.97–6.98; *p* = 0.06; I^2^ = 0%; Appendix A), increased alanine aminotransferase (OR 2.12; 95% CI 0.86–5.23; *p* = 0.10; I^2^ = 6%; Appendix A), decreased neutrophil count (OR 0.96; 95% CI 0.71–1.31; *p* = 0.81; I^2^ = 0%; Appendix A), rash (OR 4.80; 95% CI 0.85–27.00; *p* = 0.08; I^2^ = 0%; Appendix A), decreased appetite (OR 1.50; 95% CI 0.20–11.15; *p* = 0.69; I^2^ = 52%; Appendix A).

### 3.3. Sensitivity Analyses

Subgroup analyses revealed significant differences in effect sizes attributable to AT or NAT on OS (chi^2^ = 4.78; df = 1; *p* = 0.03) and nausea (chi^2^ = 11.15; df = 1; *p* < 0.01). 

We performed a leave-one-out sensitivity analysis for all outcomes. The following changes in results were found. There was a significant difference in favor of the PD-1/PD-L1 inhibitors plus chemotherapy group for OS omitting IMpower010 trial [20]. Adverse effects showed stability in the sensitivity analysis, with minimal changes. The leave-one-out sensitivity analysis of the main outcomes is detailed in Appendix A.

### 3.4. Assessment of Risk of Bias

Figure 5A presents the detailed evaluation of each RCT included in the meta-analysis. Overall, all RCTs were found to have a low risk of bias [19,20,28,29,30,31,32]. In Figure 5B, the symmetrical distribution of comparable studies in the funnel plot indicates that there is no evidence of publication bias. No significant publication bias was detected by the Egger’s (*p* = 0.1471) and Beggs’s tests (*p* = 0.3272) for the OS outcome.

## 4. Discussion 

In this systematic review and meta-analysis of seven RCTs including 3915 patients, we compared PD-1/PD-L1 inhibitors plus chemotherapy as NAT or AT to chemotherapy alone in patients with resectable stage NSCLC. The main findings indicate that PD-1/PD-L1 inhibitors plus chemotherapy as NAT were associated with (I) a significant improvement in OS; (II) a significant improvement in EFS; and (III) a significant increase in MPR and pCR. Furthermore, in AT, PD-1/PD-L1 inhibitors plus chemotherapy were associated with (I) a significant improvement in DFS, (II) with a manageable safety profile in both therapies.

Lung cancer is the primary cause of cancer-related death on a global scale, wherein the majority are attributed to NSCLC [1,33,34]. In this scenario, it is estimated that around 20–30% of NSCLC patients with stage I, 50% with stage II, and 60% with stage III-A die within five years [7,35,36]. The therapeutic approach with curative potential for these cases is surgical resection, providing significant benefits to patients in stages I and II of NSCLC. Additionally, a substantial improvement is observed in stage II when adjuvant chemotherapy is administered [37]. Meanwhile, in patients with locally advanced stage NSCLC (III-A), neoadjuvant chemotherapy followed by surgical resection has been the standard treatment, which may be complemented by adjuvant chemotherapy and thoracic radiotherapy in selected cases [38,39]. However, despite acting on systemic micrometastatic disease, the effect of adjuvant chemotherapy on OS remains modest, with a benefit of only 5.4% over five years [40]. 

Preoperative strategies involving NAT have been extensively investigated with the primary objectives of downstaging the tumor prior to surgery. This approach aims to facilitate the implementation of minimally invasive surgical procedures, suppress the early development of micrometastases, reduce the likelihood of systemic relapse, and ultimately enhance overall patient survival [35,41,42]. In this context, immunotherapy stands as a remarkable therapeutic advancement that has significantly impacted patient survival in lung cancer, especially in the context of NSCLC [41]. The elucidation of knowledge about immune mechanisms and oncogenic pathways involved in NSCLC allowed the development of new immunotherapeutic modalities [41,43,44]. Currently, PD-1 and PD-L1 inhibitors have gained approval as the first-line therapy for metastatic NSCLC patients, consistently demonstrating better results for OS and PFS [5,39]. As a result, there is a growing focus on exploring the potential of immunotherapy as a curative treatment for early-stage NSCLC [39].

NAT serves an alternative approach for managing patients with operable NSCLC and is worth considering for those with borderline resectable NSCLC [36]. In the EMERGING-CTONG 1103 trial, a NAT strategy with tyrosine kinase inhibitor was employed in patients harboring EGFR sensitivity mutations and R0-resected stage IIIA-N2 disease [45]. The group treated with Erlotinib demonstrated improved PFS compared to the group treated with gemcitabine plus cisplatin [45]. This neoadjuvant approach with anti-PD-1/PD-L1 plus antibody to cytotoxic T-lymphocyte antigen-4 (anti-CTLA-4) in the NEOSTAR trial demonstrated an improvement in pCR [46]. Hence, when specifically examining immunotherapy in the NAT setting, initial findings indicate the possibility of a prospective paradigm shift in therapy [36]. Corroborating this projection, the positive pCR rate presented in the NADIM II trial for the NAT subgroup with chemotherapy and nivolumab was accompanied by an improved OS [30]. Furthermore, the upgrade in OS was also perceived in the CheckMate 816 and KEYNOTE-671 trials [19,28]. These analyses, also applied to the EFS outcome, were confirmed in our meta-analysis, with significant results in favor of the NAT with anti PD-1/PD-L1 agents [19,28,31].

In this context, although immunotherapy for NSCLC resectable cases has been approved by the Food and Drug Administration (FDA) for both AT and NAT [47,48], AT has not always been linked to significant outcomes [20,28]. This can be seen in IMpower010 and KEYNOTE-671 trials, which presented significant improvements in DFS, but non-significant outcomes concerning the OS [20,28]. Additionally, the overall response rate in the IMpower010 was not statistically significant [20]. In our meta-analysis, similar results were encountered, with a significant improvement in the DFS and no significance in the OS outcome, when evaluating the addition of PD-1/PD-L1 inhibitors to chemotherapy in AT.

Furthermore, when analyzing subgroup treatments, the KEYNOTE-671 trial, which performed a neoadjuvant immunotherapy treatment, described a beneficial association between Pembrolizumab efficacy and the PD-L1 tumor expression. [28]. This result was remarkably confirmed in our meta-analysis, indicating a significant reduction in disease progression, disease recurrence, or death in individuals with PD-L1 tumor proportion score of ≥50% [28]. In addition, the ADAURA trial evaluated osirmetinib action on an adjuvant setting for EGFR-mutated NSCLC [49]. Its final analysis reported a significant five-year OS of 88% in the Osimertinib group versus 78% in the placebo group [49]. Moreover, in a recent trial, the adjuvant Osimertinib efficacy has been evaluated for EGFR mutations in NSCLC in association with cisplatin plus vinorelbine doublet-chemotherapy, providing valuable insights into the treatment landscape for NSCLC [50]. 

Regarding ALK mutations in NSCLC cases, the effectiveness of neoadjuvant crizotinib has been documented in 11 pathologically confirmed N2 ALK+ patients, of whom 91% underwent R0 resections and two presented pathologic complete responses [51]. Furthermore, a previous study reported the achievement of a significant pathological response with neoadjuvant alectinib in a patient diagnosed with stage IIIA ALK+ NSCLC [52]. This remarkable finding acted as a catalyst for the development of the current ALENO trial, which aims to investigate the activity and efficacy of alectinib as NAT in surgically resectable stage III ALK+ NSCLC [52]. In this context, there has been significant progress in the development of immunotherapies for different types of malignancies, considering the varied responses and the presence of mutations [44,53,54].

This meta-analysis showed no association for the addition of anti-PD-1/PD-L1 therapy to chemotherapy with severe toxicities (grade ≥ 3), compared to the chemotherapy alone group. Our results support that the addition of immunotherapy is associated with increased mild adverse events. Immune-mediated events, correlated to the anti-PD1/PD-L1 group, such as hypothyroidism and rash, were also present. These immune-related adverse events have already been described in the literature and are a consequence of the storm of inflammatory cytokines triggered by ICI when they activate the immune system [55,56,57,58]. This immune activation can lead to an attack on normal organs, resulting in a variety of toxic side effects [55,56,57,58]. We emphasize that all these events were manageable, thus showing the safety of the therapy.

## 5. Limitations

This study has limitations. First, there was moderate-to-high heterogeneity in some of the outcomes analyzed. Second, the RCTs included in this analysis involve various PD-1/PD-L1 inhibitors and chemotherapy regimens. To address this heterogeneity and assess the stability of our results, we conducted a sensitivity analysis, taking into account the different treatment regimens and potential effects of the individual studies. Third, it was not possible to perform subgroup analysis for each stage of NSCLC or types of mutations, due to studies reporting non-meaningful outcomes for each subgroup of patients. Fourth, we were unable to conduct detailed analyses on AT regimens, mostly due to only two RCTs being conducted with AT, although there is a satisfactory population.

## 6. Conclusions

In this comprehensive systematic review and meta-analysis of RCTs, the incorporation of PD-1/PD-L1 inhibitors alongside chemotherapy offers a promising prospect for reshaping the established treatment paradigms for patients diagnosed with resectable stages of NSCLC. Moreover, our analyses support the idea that neoadjuvant administration with these agents should be encouraged, in light of the fact that it was associated with an increased survival and pathological response, at the expense of a manageable safety profile.

## Figures and Tables

**Figure 1 cancers-15-05143-f001:**
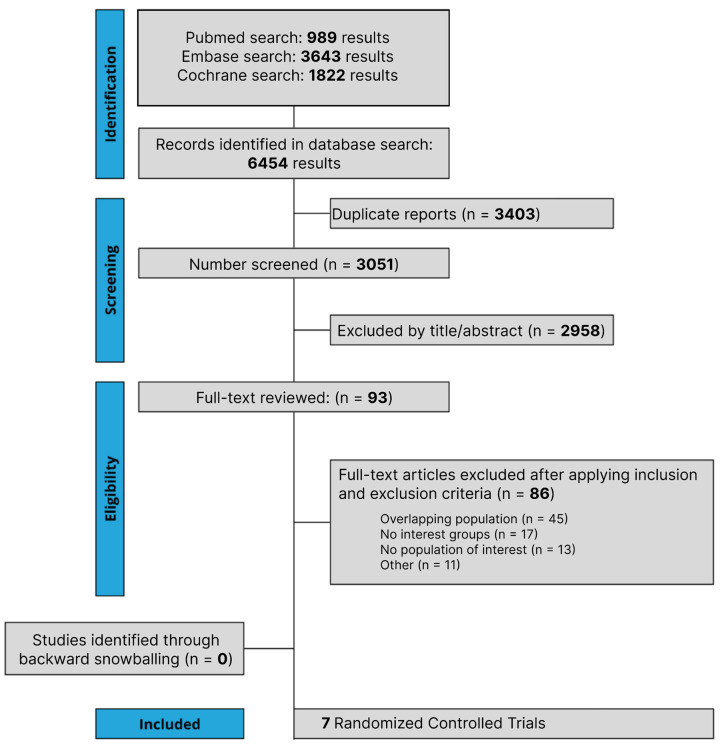
Preferred Reporting Items for Systematic Review and Meta-Analysis (PRISMA) flow diagram of study screening and selection. The search strategy in PubMed, Cochrane Library, and Embase yielded 6454 studies, of which 93 were fully reviewed for inclusion and exclusion criteria. Seven studies were included in the meta-analysis.

**Figure 2 cancers-15-05143-f002:**
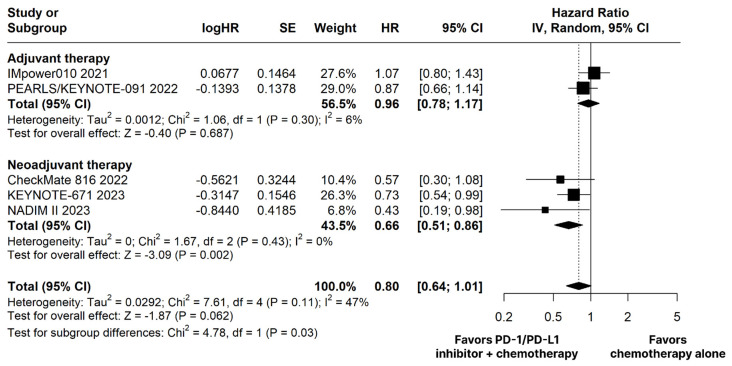
Overall survival (OS) of patients with resectable stage non-small cell lung cancer treated with programmed cell death protein 1 (PD−1)/programmed death-ligand 1 (PD−L1) inhibitors plus chemotherapy versus chemotherapy alone. CI, confidence interval; HR, hazard ratio; IV, inverse variance; SE, standard error.

**Figure 3 cancers-15-05143-f003:**
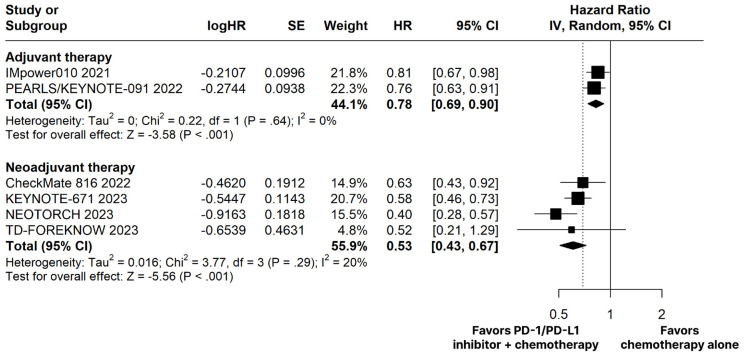
Disease-free survival (DFS) in adjuvant therapy and event-free survival (EFS) in neoadjuvant therapy of patients with resectable stage non-small cell lung cancer treated with programmed cell death protein 1 (PD−1)/programmed death-ligand 1 (PD−L1) inhibitors plus chemotherapy versus chemotherapy alone. CI, confidence interval; HR, hazard ratio; IV, inverse variance; SE, standard error.

**Figure 4 cancers-15-05143-f004:**
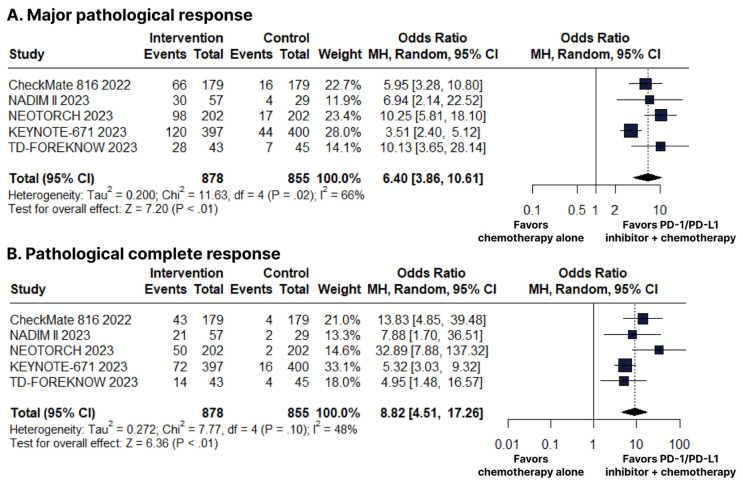
(**A**) Major pathological response (MPR). (**B**) Pathological complete response (pCR). Comparison between programmed cell death protein 1 (PD−1)/programmed death-ligand 1 (PD−L1) inhibitors plus chemotherapy versus chemotherapy alone in patients with resectable stage non-small cell lung cancer. CI, confidence interval; MH, Mantel–Haenszel.

**Figure 5 cancers-15-05143-f005:**
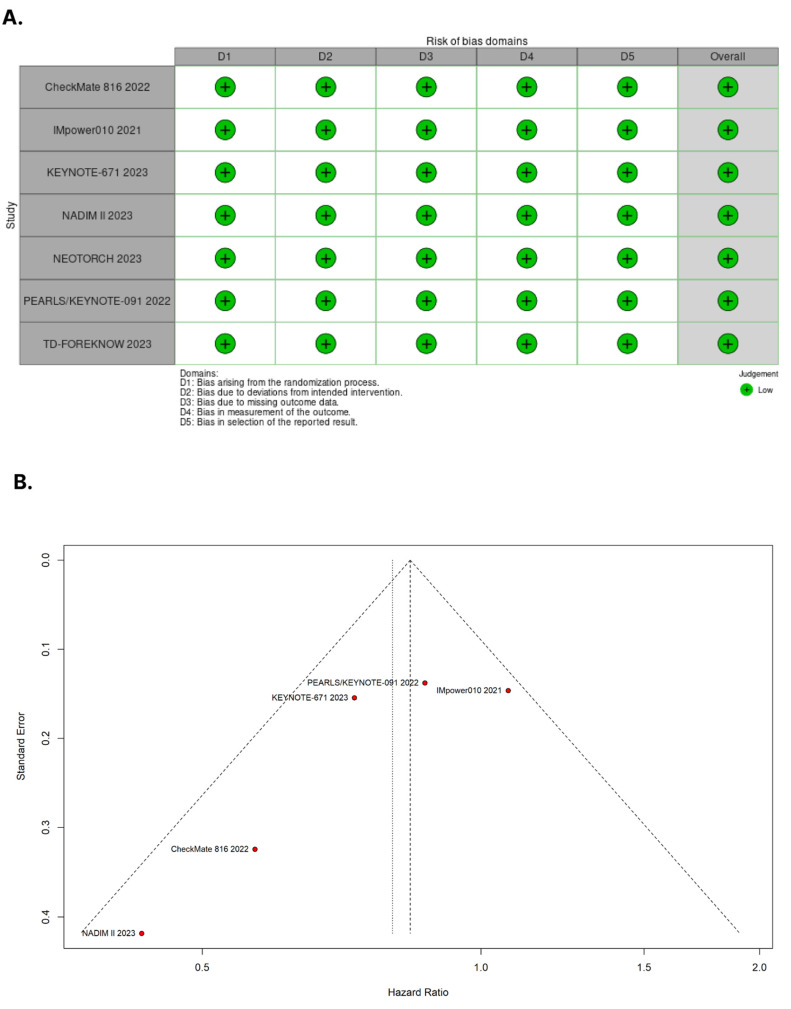
(**A**) Critical appraisal of randomized controlled trials according to the Cochrane Collaboration’s tool for assessing risk of bias in randomized trials. (**B**) Funnel plot analysis of the overall survival of patients with non-small cell lung cancer shows no evidence of publication bias.

**Table 1 cancers-15-05143-t001:** Design and characteristics of studies included in the meta-analysis.

Study	Design/NCT	Intervention Treatment	Follow-Up (Median)	Number of Participants IG/CG, No.	Median Age (Range or IQR) IG/CG, Years	Male IG/CG, No. (%)	PD-L1 Status ≥ 50% IG/CG, No. (%)	Histology, No. (%)	
								IG	CG
Adjuvant therapy									
IMpower010 2021	RCT—phase III/NCT02486718	Atezolizumab 1200 mg	32.2 months	507/498	62 (IQR, 57–67)/62 (IQR, 56–68)	337 (66)/335 (67)	NA/NA	328 (65) Non-squamous, 179 (35) Squamous	331 (67) Non-squamous, 167 (34) Squamous
PEARLS/KEYNOTE-091 2022	RCT—phase II/NCT02504372	Pembrolizumab 200 mg	35.6 months	590/587	65 (IQR, 59–70)/65 (IQR, 59–70)	401 (68)/403 (69)	168 (28)/165 (28)	398 (67) Non-squamous, 192 (33) Squamous	363 (62) Non-squamous, 224 (38) Squamous
Neoadjuvant therapy									
CheckMate 816 2022	RCT—phase III/NCT02998528	Nivolumab 360 mg	29.5 months	179/179	64 (range, 41–82)/65 (range, 34–84)	128 (71.5)/127 (70.9)	38 (21.2)/42 (23.5)	92 (51.4) Non-squamous, 87 (48.6) Squamous	84 (46.9) Non-squamous, 95 (53.1) Squamous
KEYNOTE-671 2023	RCT—phase III/NCT03425643	Pembrolizumab 200 mg	25.2 months	397/400	63 (range, 26–83)/64 (range, 35–81)	279 (70.3)/284 (71.0)	132 (33.2)/134 (33.5)	226 (56.9) Non-squamous, 171 (43.1) Squamous	227 (56.8) Non-squamous, 173 (43.2) Squamous
NADIM II 2023	RCT—phase II/NCT03838159	Nivolumab 360 mg	26.1 months	57/29	65 (IQR, 58–70)/63 (IQR, 57–66)	36 (63)/16 (55)	NA/NA	36 (63) Non-squamous *, (37) Squamous-cell carcinoma	15 (52) Non-squamous *, 14 (48) Squamous-cell carcinoma
NEOTORCH 2023	RCT—phase III/NCT04158440	Toripalimab 240 mg	18.3 months	202/202	NA/NA	NA/NA	NA/NA	202 (100) Non-squamous	202 (100) Non-squamous
TD-FOREKNOW 2023	RCT—phase II/NCT04338620	Camrelizumab 200 mg	14.1 months	43/45	61 (IQR 54–65)/61 (IQR 54–65)	34 (79.1)/40 (88.9)	NA/NA	16 (37.2) Non-squamous *, 27 (62.8) Squamous-cell carcinoma	13 (28.8) Non-squamous *, 32 (71.1) Squamous-cell carcinoma

* Included: adenocarcinoma, large-cell carcinoma, not otherwise specified or undifferentiated or other. CG, control group; IG, intervention group; IQR, interquartile range; NA, not available; NCT, National Clinical Trial; RCT, randomized controlled trial; PD-L1, programmed death-ligand 1.

## Data Availability

The data for this study were systematically collected and organized into a comprehensive database. Access to the data can be granted upon request from the corresponding author.

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
