# Peer review of "PD-1/PD-L1 Inhibitors plus Chemotherapy Versus Chemotherapy Alone for Resectable Non-Small Cell Lung Cancer: A Systematic Review and Meta-Analysis of Randomized Controlled Trials"

_cancers, 2023, doi:10.3390/cancers15215143_

Round 1
Reviewer 1 Report
The manuscript entitled "PD-1/PD-L1 Inhibitors Plus Chemotherapy Versus Chemotherapy Alone for Resectable Non-Small Cell Lung Cancer: A Systematic Review And Meta-Analysis of Randomized Controlled 4 Trials" is a comprehensive systematic review on PD1/PDL1 inhibitors based on the analysis of randomized controlled clinical trials. This study carefully analysed over 3000 references from various sources such as PubMed, Embase and Cochrane. The inclusion and exclusion criteria for the study is carefully chosen and selected based on PRISMA guidelines. This analysis reported that the integration of PD-1/PD-L1 inhibitors in conjunction with chemotherapy presents a promising avenue for the reconfiguration of established treatment protocols for individuals afflicted with stages of non-small cell lung cancer (NSCLC). Furthermore, our analyses substantiate the recommendation of neoadjuvant application of these agents, given its association with enhanced survival rates and favorable pathological responses, all within the confines of a manageable safety profile.
All the results are well explained with the help of tables and figures. The idea is novel and no previous reports are available in this subject.
Hence, I recommend this manuscript for the publication in journal ‘Cancers’.
Author Response
RESPONSE TO REVIEWER #1
The manuscript entitled "PD-1/PD-L1 Inhibitors Plus Chemotherapy Versus Chemotherapy Alone for Resectable Non-Small Cell Lung Cancer: A Systematic Review And Meta-Analysis of Randomized Controlled 4 Trials" is a comprehensive systematic review on PD1/PDL1 inhibitors based on the analysis of randomized controlled clinical trials. This study carefully analysed over 3000 references from various sources such as PubMed, Embase and Cochrane. The inclusion and exclusion criteria for the study is carefully chosen and selected based on PRISMA guidelines. This analysis reported that the integration of PD-1/PD-L1 inhibitors in conjunction with chemotherapy presents a promising avenue for the reconfiguration of established treatment protocols for individuals afflicted with stages of non-small cell lung cancer (NSCLC). Furthermore, our analyses substantiate the recommendation of neoadjuvant application of these agents, given its association with enhanced survival rates and favorable pathological responses, all within the confines of a manageable safety profile.
All the results are well explained with the help of tables and figures. The idea is novel and no previous reports are available in this subject.
Hence, I recommend this manuscript for the publication in journal ‘Cancers’.
In response: We thank the reviewer for the thoughtful consideration of our manuscript and recommend it for publication.
Reviewer 2 Report
ABSTRACT OK
INTRODUCTION OK
METHODS OK
RESULTS OK
DISCUSSION SHOULD BE MORE FOCUSED WITH LESS WORDS
LIMITATIONS MANY !!!ARE YOU SURE YOU CANT DO ANYTHING BETTER?
CONCLUSIONS OK
Author Response
RESPONSE TO REVIEWER #2
1. ABSTRACT OK
In response: We thank the reviewer for the thoughtful consideration of our manuscript.
2. INTRODUCTION OK
In response: We thank the reviewer for the thoughtful consideration of our manuscript.
3. METHODS OK
In response: We thank the reviewer for the thoughtful consideration of our manuscript.
4. RESULTS OK
In response: We thank the reviewer for the thoughtful consideration of our manuscript.
5. DISCUSSION SHOULD BE MORE FOCUSED WITH LESS WORDS
In response: We thank the reviewer for this comment. Our paper is the first systematic review and meta-analysis evaluating the efficacy and safety of neoadjuvant or adjuvant use of PD-1/PD-L1 inhibitors plus chemotherapy versus chemotherapy alone in resectable stage (I-III) non-small cell lung cancer (NSCLC). Therefore, our objective was to broadly review the literature on treatments for resectable stage NSCLC, which made our discussion a little longer than usual. However, we believe that the presentation of this overview of treatments for NSCLC is necessary in view of the constant research on the topic. Currently, immunotherapy has proven to be very important and effective in the treatment of different tumors, however, it is important to consider that the presence of mutations requires specific assessments, therefore, we seek to synthetically present these different therapies in relation to NSCLC in our discussion, which should be considered in the clinical assessment of patients.
6. LIMITATIONS MANY !!!ARE YOU SURE YOU CAN DO ANYTHING BETTER?
In response: We thank the reviewer for this comment. We consider it essential for clinical practice to understand the limitations of our meta-analysis, given the need for more studies to clarify topics that were not possible to analyze, such as the different stages of NSCLC and types of mutations. In addition, only two randomized clinical trials have been published evaluating the adjuvant therapy of PD-1/PD-L1 inhibitors plus chemotherapy versus chemotherapy in patients with resectable stage NSCLC. In this context, we highlight the need to maintain the limitations of our meta-analysis, serving as a basis for directing future research.
7. CONCLUSIONS OK
In response: We thank the reviewer for the thoughtful consideration of our manuscript.
Reviewer 3 Report
In their manuscript, Pasqualotto and collaborators reviewed and performed a meta-analysis comparing two therapeutic regimen: PD-1/PD-L1 inhibitors + chemotherapy vs chemotherapy alone in resectable NSCLC. The authors found that combining CIs with NAT significantly improved OS and EFS compared to chemotherapy alone. They further found that there was a significant increase in favor of CIs + chemo for major pathological response and pathological complete response. However, the authors found that ICs combined with AT was significantly beneficial for DFS but not OS.
The manuscript is clearly written and contains a lot of detailed figures, tables and supplemental figures to support the authors’ statements. Therefore, no change is needed.
Author Response
RESPONSE TO REVIEWER #3
In their manuscript, Pasqualotto and collaborators reviewed and performed a meta-analysis comparing two therapeutic regimen: PD-1/PD-L1 inhibitors + chemotherapy vs chemotherapy alone in resectable NSCLC. The authors found that combining CIs with NAT significantly improved OS and EFS compared to chemotherapy alone. They further found that there was a significant increase in favor of CIs + chemo for major pathological response and pathological complete response. However, the authors found that ICs combined with AT was significantly beneficial for DFS but not OS.
The manuscript is clearly written and contains a lot of detailed figures, tables and supplemental figures to support the authors’ statements. Therefore, no change is needed.
In response: We thank the reviewer for the thoughtful consideration of our manuscript.
Reviewer 4 Report
Dear authors
The manuscript is well-written, in a clear and concise manner.
The material and methods are very robust and allow authors to achieve the proposed aim.
Some small improvements are suggested to authors:
Al Latin words must be in italic, such as versus and its abbreviation vs.
The abbreviations must be first written fully and only after brackets the abbreviation, and from there on, just use the abbreviations. Abbreviations do not have plural, check all manuscript.
Table titles and figure legends must be very detailed and descriptive of the content of the Figure and Table.
All abbreviations used on tables and figures must have an abbreviation legend for each figure or table.
Nothing to add
Author Response
RESPONSE TO REVIEWER #4
1. The manuscript is well-written, in a clear and concise manner.
The material and methods are very robust and allow authors to achieve the proposed aim.
In response: We thank the reviewer for the thoughtful consideration of our manuscript. We believe it is improved as a result of your suggestions, as outlined below.
2. Al Latin words must be in italic, such as versus and its abbreviation vs.
In response: We thank the reviewer for the thoughtful suggestion. We corrected the manuscript and the modifications have been highlighted.
3. The abbreviations must be first written fully and only after brackets the abbreviation, and from there on, just use the abbreviations. Abbreviations do not have plural, check all manuscript.
In response: We thank the reviewer for bringing this to our attention. We corrected the manuscript and the modifications have been highlighted.
4. Table titles and figure legends must be very detailed and descriptive of the content of the Figure and Table.
In response: We thank the reviewer for this comment. We reviewed the figures and tables in the manuscript and provided more details as needed for understanding. The modifications have been highlighted.
5. All abbreviations used on tables and figures must have an abbreviation legend for each figure or table.
In response: We thank the reviewer for this suggestion. We have updated the legend for each figure and table including the abbreviations used. The modifications have been highlighted.
Reviewer 5 Report
The authors conducted a systematic review and meta-analysis to evaluate the benefit of adding PD-1/PD-L1 inhibitors to the treatment of early-stage non-small cell lung cancer (NSCLC), as both neoadjuvant therapy (NAT) and adjuvant therapy (AT). They searched PubMed, Embase, and Cochrane databases for randomized controlled trials (RCTs) investigating PD-1/PD-L1 inhibitors plus chemotherapy in resectable NSCLC.
A total of seven RCTs with 3,915 patients were identified that randomized patients to chemotherapy with or without PD-1/PD-L1 inhibitors as NAT or AT. As NAT, PD-1/PD-L1 inhibitors plus chemotherapy significantly improved overall survival and event-free survival compared to chemotherapy alone. There were also significant increases in major pathological response and pathological complete response rates.
As AT, disease-free survival favored the PD-1/PD-L1 inhibitor combination group. The safety profiles were manageable. The meta-analysis provides supportive evidence that incorporating PD-1/PD-L1 inhibitors with chemotherapy can reshape NAT and AT paradigms for resectable NSCLC. Neoadjuvant use, associated with increased survival and pathological responses, was recommended.
In summary, the review demonstrated potential benefits of PD-1/PD-L1 inhibitors in the neoadjuvant and adjuvant settings for early-stage NSCLC based on available RCT data.
Apparently, incorporating PD-1/PD-L1 inhibitors with chemotherapy in resectable stages of NSCLC, particularly in the neoadjuvant setting, is crucial. Such findings underscore the importance of understanding and leveraging molecular mechanisms for optimized treatment strategies. For instance, recent investigations into SCLC have revealed the tumorigenic contribution of certain transcripts like circPVT1 and chimPVT1, suggesting a possible functional connection between MYC and YAP1/POU2F3. Such molecular insights may, in the future, not only serve as novel biomarkers but could also direct more individualized treatment regimens. As the landscape of lung cancer treatment continues to evolve, it is imperative that we deepen our molecular understanding, similar to advances in SCLC, to reshape established treatment paradigms and improve patient outcomes (please refer to doi.org/10.1002/gcc.23121 and expand the introduction and discussion sections).
The authors conducted a systematic review and meta-analysis to evaluate the benefit of adding PD-1/PD-L1 inhibitors to the treatment of early-stage non-small cell lung cancer (NSCLC), as both neoadjuvant therapy (NAT) and adjuvant therapy (AT). They searched PubMed, Embase, and Cochrane databases for randomized controlled trials (RCTs) investigating PD-1/PD-L1 inhibitors plus chemotherapy in resectable NSCLC.
A total of seven RCTs with 3,915 patients were identified that randomized patients to chemotherapy with or without PD-1/PD-L1 inhibitors as NAT or AT. As NAT, PD-1/PD-L1 inhibitors plus chemotherapy significantly improved overall survival and event-free survival compared to chemotherapy alone. There were also significant increases in major pathological response and pathological complete response rates.
As AT, disease-free survival favored the PD-1/PD-L1 inhibitor combination group. The safety profiles were manageable. The meta-analysis provides supportive evidence that incorporating PD-1/PD-L1 inhibitors with chemotherapy can reshape NAT and AT paradigms for resectable NSCLC. Neoadjuvant use, associated with increased survival and pathological responses, was recommended.
In summary, the review demonstrated potential benefits of PD-1/PD-L1 inhibitors in the neoadjuvant and adjuvant settings for early-stage NSCLC based on available RCT data.
Apparently, incorporating PD-1/PD-L1 inhibitors with chemotherapy in resectable stages of NSCLC, particularly in the neoadjuvant setting, is crucial. Such findings underscore the importance of understanding and leveraging molecular mechanisms for optimized treatment strategies. For instance, recent investigations into SCLC have revealed the tumorigenic contribution of certain transcripts like circPVT1 and chimPVT1, suggesting a possible functional connection between MYC and YAP1/POU2F3. Such molecular insights may, in the future, not only serve as novel biomarkers but could also direct more individualized treatment regimens. As the landscape of lung cancer treatment continues to evolve, it is imperative that we deepen our molecular understanding, similar to advances in SCLC, to reshape established treatment paradigms and improve patient outcomes (please refer to doi.org/10.1002/gcc.23121 and expand the introduction and discussion sections).
Author Response
RESPONSE TO REVIEWER #5
1. The authors conducted a systematic review and meta-analysis to evaluate the benefit of adding PD-1/PD-L1 inhibitors to the treatment of early-stage non-small cell lung cancer (NSCLC), as both neoadjuvant therapy (NAT) and adjuvant therapy (AT). They searched PubMed, Embase, and Cochrane databases for randomized controlled trials (RCTs) investigating PD-1/PD-L1 inhibitors plus chemotherapy in resectable NSCLC.
A total of seven RCTs with 3,915 patients were identified that randomized patients to chemotherapy with or without PD-1/PD-L1 inhibitors as NAT or AT. As NAT, PD-1/PD-L1 inhibitors plus chemotherapy significantly improved overall survival and event-free survival compared to chemotherapy alone. There were also significant increases in major pathological response and pathological complete response rates.
As AT, disease-free survival favored the PD-1/PD-L1 inhibitor combination group. The safety profiles were manageable. The meta-analysis provides supportive evidence that incorporating PD-1/PD-L1 inhibitors with chemotherapy can reshape NAT and AT paradigms for resectable NSCLC. Neoadjuvant use, associated with increased survival and pathological responses, was recommended.
In summary, the review demonstrated potential benefits of PD-1/PD-L1 inhibitors in the neoadjuvant and adjuvant settings for early-stage NSCLC based on available RCT data.
In response: We thank the reviewer for the thoughtful consideration of our manuscript. We believe it is improved as a result of your suggestions, as outlined below.
2. Apparently, incorporating PD-1/PD-L1 inhibitors with chemotherapy in resectable stages of NSCLC, particularly in the neoadjuvant setting, is crucial. Such findings underscore the importance of understanding and leveraging molecular mechanisms for optimized treatment strategies. For instance, recent investigations into SCLC have revealed the tumorigenic contribution of certain transcripts like circPVT1 and chimPVT1, suggesting a possible functional connection between MYC and YAP1/POU2F3. Such molecular insights may, in the future, not only serve as novel biomarkers but could also direct more individualized treatment regimens. As the landscape of lung cancer treatment continues to evolve, it is imperative that we deepen our molecular understanding, similar to advances in SCLC, to reshape established treatment paradigms and improve patient outcomes (please refer to doi.org/10.1002/gcc.23121 and expand the introduction and discussion sections).
In response: We thank the reviewer for this suggestion. We agree with the reviewer about the importance of molecular differences in malignancies in the development of immunotherapies, so we added this topic to the manuscript discussion. We also reference the study by Tolomeo et al. as requested by the reviewer, in addition to the studies by Abbott et al. and Rolfo et al. The modifications have been highlighted.
Round 2
Reviewer 5 Report
The authors have clarified several of the questions I raised in my previous review. Most of the major problems have Been addressed by this revision.
The authors have clarified several of the questions I raised in my previous review. Most of the major problems have Been addressed by this revision.